# Oportuna Vacuna: A Prospective Study of Vaccine Confidence and Vaccine Uptake in a Low-Income, Spanish-Speaking Rhode Island Population in the Post-Pandemic Era

**DOI:** 10.3390/vaccines14010002

**Published:** 2025-12-19

**Authors:** Julia Testa, Morgan Leonard, Chilsea J. Wang, Jaqueline Medrano, Sharon Farrar, Anne Searls De Groot

**Affiliations:** 1Clinica Esperanza/Hope Clinic, Providence, RI 02909, USAjackiem@aplacetobehealthy.org (J.M.);; 2School of Medicine, University of California, Davis Campus, Sacramento, CA 95817, USA; chilseajwang@gmail.com; 3EpiVax Inc., Providence, RI 02909, USA

**Keywords:** vaccine hesitancy, uninsured, free clinic, hispanic/latino, community health workers, vaccine acceptance

## Abstract

**Background:** Clínica Esperanza/Hope Clinic (CEHC), a free clinic for low-income, uninsured, Spanish-speaking patients, located in Providence, RI, piloted an intervention to improve vaccination rates. This program, named “Oportuna Vacuna” (OV) aimed to assess vaccine hesitancy in the post-pandemic period and measure improvements to vaccine uptake after integrating healthcare provider (HCP) and patient education with vaccine workflow adjustments. **Methods:** OV was initiated in January 2023. Culturally attuned and linguistically appropriate vaccine-focused educational programs were developed and provided to patients and HCPs, while workflow modifications to streamline vaccine administration were implemented during clinic visits. Structured pre- and post-intervention chart reviews were conducted and (oral- and) paper-based knowledge, attitudes, and practices surveys were administered to selected staff and patients to assess knowledge, attitudes, and vaccine confidence before and after the interventions. A total of 816 charts were reviewed prior to the intervention; 709 charts were reviewed post-intervention. A total of 72 patient and 10 HCP pre- and post- intervention surveys were completed. Vaccine uptake was compared to the uptake in 2022. **Results:** Overall vaccination rates at the clinic increased by more than 400% compared to the prior year. Patients and HCPs demonstrated increased vaccine knowledge and confidence, particularly with respect to COVID-19 and HPV vaccines. In contrast, chart reviews of individual patients revealed only a slight improvement in vaccination completion rates for patients over the 1-year period. **Conclusion:** Substantial increases in vaccine administration rates across multiple vaccine types highlight the effectiveness of OV, mainly for first time patients, at CEHC. Clinic workflow modifications improved vaccination efficiency and increased vaccine uptake. Educational sessions on vaccines were well received by patients and staff. Overall knowledge about vaccines improved during the intervention among staff and HCPs. Patients reported higher trust in HCPs compared to other sources for vaccine information.

## 1. Introduction

Vaccine coverage is declining both in the United States and globally [1]. COVID-19 vaccine uptake became highly controversial in the US during the COVID-19 pandemic and uptake of routine vaccines was also impacted. Following the pandemic, vaccine confidence has been at an all-time low [2,3]. In parallel, the incidence of vaccine-preventable diseases such as mumps, measles, hepatitis A, and whooping cough (pertussis) is on the rise. Limited access to clinical care for some vulnerable populations, such as immigrants, uninsured, underinsured, and low-income, has further reduced the uptake of childhood vaccines [4].

Clínica Esperanza/Hope Clinic (CEHC) is a free clinic located on the west side of Providence, Rhode Island, that has extensive experience working with low-income, predominantly Spanish-speaking immigrants who live in Rhode Island. CEHC provides free access to vaccines and clinical care at two locations in the Valley neighborhood of Providence. At the main clinic, patients are seen by providers in-person or via telemedicine. Vaccine clinics are held at the Neighborhood Health Station (NHS), a multipurpose building where educational programs and health screening events are held. CEHC provides several educational and peer-support programs designed for the Spanish-speaking clinic population, including Vida Sana [5,6] and Vida Pura for patients who have substance use disorders. The NHS also supports Health Fairs where basic point-of-care tests, including blood glucose, A1C, blood pressure, and more are available to all. Both the main clinic and the NHS provided vaccine education and administration to patients (Effective February 2023, the NHS location was relocated to the main clinic and continues to provide a space for these activities).

The Oportuna Vacuna (OV) pilot intervention was designed to assess barriers to vaccination and to improve vaccine uptake over a 24-month period. The intervention involved targeted education for healthcare providers as well as patients and workflow improvements. We hypothesized that vaccine knowledge and confidence would be low among patients and staff and that targeted education efforts would increase vaccine knowledge, confidence, and uptake among both groups. Workflow improvements included focused chart review for each scheduled clinic visit (including new patient visits), informing patients about vaccines that would be given at their visit, and establishment of protocols enabling Navegantes, bilingual community health workers cross-trained as medical interpreters and medical assistants, to administer vaccines. Chart reviews before and after the intervention period were used to evaluate the impact of the intervention on vaccine uptake for previously registered patients over the 24-month period. Vaccine uptake was measured for the overall patient panel by comparing vaccinations given prior to the intervention and after the intervention. Surveys were performed before and after educational sessions for HCPs and patients to evaluate changes to knowledge, attitudes, and vaccine acceptance.

This brief intervention had a strong impact on vaccine uptake, which could be attributed to the educational sessions and workflow improvements. The primary impact appeared to be on vaccine uptake by new and returning patients, patients who returned after being out-of-state or out of the country for longer than a year, rather than improved vaccination rates for individual patients. This difference may be attributed to low overall annual follow-up rates for individual patients at CEHC, where the average visit frequency is 1.5 visits per year. Thus, many patients did not return for a second visit during the evaluation period and did not receive updated vaccines.

This program demonstrated that integrating vaccine assessments and administering vaccines as-needed at each individual patient visit, regardless of the reason for the visit, appears to be a successful strategy for transient populations such as the one accessing healthcare at a typical free clinic. Ambulatory clinic workflow interventions that prioritize vaccination, in conjunction with HCP and patient education, may be useful and easily implemented approaches for improving vaccine uptake in vulnerable populations.

## 2. Materials and Methods

This was a quality improvement study utilizing a pre- and post-intervention design. The data were collected from patient chart reviews and surveys administered to both staff and patients before and after the intervention. CEHC serves a largely uninsured, low-income immigrant population. Participants in this intervention included healthcare providers (HCPs) of various types: physicians, nurses, community health workers (CHWs), and administrative staff. Patient participants included individuals engaged in routine clinic visits, Health Fair attendees, or participants in the clinic’s lifestyle education programs, Vida Sana and Vida Pura. No artificial intelligence, generative or otherwise, was used in the creation, implementation, or analysis of this project.

### 2.1. Informed Consent for Participation

Two types of participant consent were used for this intervention. For patient chart reviews, charts were selected randomly for patients who had visited during specified time periods. Charts were reviewed by clinic personnel using a vaccine uptake tracking form. Charts were only reviewed if a signed CEHC partnership form was included in the patient documents within the chart (Appendix A). The CEHC partnership form permits CEHC personnel to track and report patient health outcomes in de-identified individual and aggregated data sets, for the duration of the individual’s participation in clinical care at CEHC.

In addition, a standardized informed consent process was followed for the individual patients and HCPs who participated in the OV educational sessions. Participants were informed about the OV objectives and asked whether they would be interested in filling out a pre- and post- intervention survey. Those individuals who responded positively to the idea of filling out surveys were then provided with information about the study and asked to complete an informed consent form.

All of the study information and all of the materials were provided to participants in Spanish and English, depending on the participant’s preference. For individuals who could not read, the consent form and survey were read orally to the participant in their preferred language by a trained CHW. For participants who could not write, oral consent was given, and a fingerprint was used in lieu of a written signature. Trained CHWs assisted individuals with completing the consent process and survey. Pre- and post-intervention surveys were completed anonymously and no identifiable information was recorded on the survey forms. The study protocol and surveys were reviewed prior to the initiation of the project and approved by Salus IRB (formerly Ethical and Independent Review Services).

### 2.2. Chart Review

Pre- and post-intervention chart reviews (*n* = 816 and *n* = 709, respectively) were conducted to assess vaccine uptake for individual patients over a one-year period. The initial 816 charts were pulled from patient visits between 1 October 2022 and 28 February 2023. The same charts were reviewed at the conclusion of the intervention and vaccine status was updated accordingly. The data collected from the charts on vaccines was stored in an encrypted Excel spreadsheet; a coding system was used to avoid disclosure of individual patient information. The results of the chart review were analyzed by clinic personnel. Charts were removed from the post-intervention chart review if the patient fit one or more of the following criteria: they had transferred care to another health center, moved out of Rhode Island, were deceased, or the clinic was unable to contact the patient after three phone call attempts and if no response was received to outreach letters sent to their last known address during the intervention period.

### 2.3. Patient Vaccine Education Programs

The vaccine education sessions were provided to all of the patients who were participating in health education classes at CEHC. For participants who were willing to participate in a pre- and post-education session survey, consent forms were provided and consent was obtained prior to the education session (see informed consent section). Patient education sessions were led by multilingual CHWs (Navegantes), who provided culturally sensitive education on vaccines in Spanish. Topics covered in these sessions included vaccine basics, why vaccines are an important health intervention, and detailed information about the vaccines offered at CEHC. Special sessions were devoted to COVID-19 and HPV, as both vaccines have a greater impact on health outcomes in populations that have constraints on access to specialized care. Material included in the presentation was sourced from the World Health Organization (WHO) [7], National Cancer Institute [8], and Rhode Island Department of Health (RIDOH) [9]. Participants of these sessions viewed a presentation about the aforementioned topics with an open question-and-answer and discussion session with the Navegante instructors following the conclusion of the presentation. The topics in the presentation and discussion session aligned with the topics covered in the pre- and post-intervention surveys. All class attendees received the presentation and participated in the discussion, only those who wished to participate in the study were consented following the consent process outlined in the “Informed Consent for Participation” section prior to completing the pre-survey.

### 2.4. Healthcare Provider Educational Programs

Medical providers, including volunteer medical doctors, physician assistants, nurse practitioners, and registered nurses (MDs, PAs, NPs, and RNs), also participated in vaccine education programs; however, these were individual efforts, using “continuing education” materials available online. For example, individual providers completed the CDC’s “You Call the Shots” online modules during the intervention period. These “You Call the Shots” modules addressed vaccine recommendations including those for the HPV, flu, hepatitis B, diphtheria, tetanus and pertussis (Tdap), and pneumococcal pneumonia vaccines. Other topics covered by the CDC online modules included vaccine handling, storage, best practices, and immunization schedules. Additionally, providers were asked to complete the International Pediatric Association’s (IPA) Vaccine Trust Project modules online. Topics covered in the IPA modules included “infodemiology [10]”, interpersonal communication, techniques related to talking about vaccines with vaccine deniers, and how to be a vaccine advocate. Other CME courses include “UpToDate” courses and CDC webinars on vaccines.

HCPs also participated in monthly “Lunch-and-Learn” sessions, which were held from March 2023 to December 2023, covering education materials related to vaccination guidelines and best practices. Importantly, all members of the clinic staff, including CHWs, nurses, administrative staff, and providers participated in the “Lunch-and-Learn” sessions. Topics covered included motivational interviewing and the AIMS method for achieving higher vaccine uptake, HPV vaccinations, vaccination best practices, current CDC guidelines, and more. Training materials were sourced from the CDC’s “You Call the Shots” online training module series, the WHO, the IPA’s Vaccine Trust Project online modules, and RIDOH. The sessions were presented to staff through PowerPoint presentations in English with real-time Spanish translation, as many of the CHWs at CEHC speak Spanish as their first language. Paper-based knowledge, attitudes, and practices surveys were administered to both staff and patients, pre- and post-intervention, to evaluate their knowledge and attitudes towards vaccines (Appendix A).

The AIMS approach [11] was implemented during OV and was explained to HCPs during the “Lunch-and-Learn” sessions and was provided to volunteers in an online module. AIMS is an acronym that stands for “Announce, Inquire, Mirror, Secure”. Vaccine providers are asked to Announce the plan to vaccinate during a patient’s clinic visit in a professional manner. If the vaccine recipient hesitates, the vaccine provider then Inquires about the hesitation, asking open-ended questions and seeking to understand the patient’s hesitancy or concerns. Instead of responding directly to patient’s concerns, the AIMS method recommends that the provider Mirror their response, demonstrating active listening. The Inquire–Mirror steps continue until the provider and patient establish that the HCP understands the patient’s vaccine-related concerns. Once that agreement is reached, the provider moves to Secure trust by responding to the patient’s concerns with information that fits the patient’s needs. If the patient is still hesitant or refuses, then the provider can explain that while they disagree with the patient, they share a concern for the patient’s health. The focus is on securing trust and mutual respect which then enables the HCP to provide all, or some, of the vaccines that are recommended to the patient. The vaccination then proceeds or the plan to vaccinate will be modified to fit the agreement between the patient and the HCP.

### 2.5. Clinic Workflow Changes

Clinical interventions were primarily focused on workflow improvements. For example, pre-visit chart reviews identified the vaccines that were due for each patient and the Navegantes informed patients of their vaccination needs during check-in for their appointment using the AIMS approach described above. Vaccines were administered by Navegantes and nursing staff using a pre-approved clinic protocol (based on the 2023 recommendations) [12], reducing the need for individual review and approval by supervisory HCPs. Additionally, the Navegantes added the appropriate vaccine consent forms and required VIS documents to the patient’s folder prior to the visit. Doing so ensured that all of the Navegantes and HCPs that worked with that patient were aware of the vaccines the patient was going to be receiving during the visit. A vaccination z-code was automatically added to every office visit type in the electronic medical record system (EMR), which is an added reminder for Navegantes and HCPs to discuss vaccination with every patient at every visit, as warranted.

The primary outcomes related to the clinic workflow changes included vaccination records (number of vaccines distributed at the clinic, based on nursing documentation), retrospective chart review prior to the interventions listed above and during the post-intervention period.

## 3. Results

### 3.1. Clinic Population

CEHC primarily serves Spanish-speaking immigrants that work in low-income employment, are uninsured, have multiple chronic health problems, and who have low health literacy [13]. Most of the HCP staff members are also immigrants or second-generation Americans. CEHC patients have limited access to healthcare as they are not eligible for subsidized or public medical insurance due to their recent immigration status—public support such as Medicaid is not available until the individual has been a legal resident for more than five years—and their employment income is usually insufficient to pay for employer-based or private health insurance.

### 3.2. Chart Review Population

A total of 816 patient charts were randomly selected for the initial chart review (CR1); 709 of these individuals were still active patients at CEHC at the time of the post-intervention chart review (CR2). Of the patients included in the chart review, 48% identified as female and 52% identified as male, with 0.1% identifying as transgender; these proportions did not change between CR1 and CR2. The ages of the patients ranged from 18 to 87 years old and the average and median age of both CR populations was 47 years old. Approximately 93% of the patients included in the chart review identified as Hispanic or Latino, 5% identified as Black/African American, and 14% identified as White.

### 3.3. Survey Participants

Of the 36 individuals who completed surveys pre-and post-educational sessions, 69% were female, 25% were male, and 0.5% chose not to disclose their gender. Ages ranged from 23 to 62 years old. Survey respondents were also asked about their education history: 19% reported that they had only completed primary school, 42% reported completing high school, and 25% had completed college. Staff surveys (10 total) were also performed.

### 3.4. Chart Review Results

The vaccine uptake and completion status for the individual patients evaluated during chart review I (CR1) and chart review II (CR2) improved, but not at the same pace as the overall vaccination rate at CEHC. The patient population at CEHC is highly transient, and patients often move unexpectedly, transferring care to other health centers when they become insured, change jobs, or are otherwise lost to follow-up. A total of 107 (13%) charts were removed from the second chart review (CR2) and for those reasons, a total of 709 charts were reviewed in CR2.

Chart review focused on the following vaccines and used the CDC recommendations for vaccination for the age and health status of the individual: flu, Tdap, pneumococcal pneumonia, hepatitis B, measles, mumps, rubella (MMR), varicella, HPV, and COVID-19. For the purpose of this study, “up to date”, for COVID-19 vaccines, was defined as having received a full series of Johnson & Johnson (one-dose series) or Pfizer or Moderna (two-dose series) vaccinations and a booster (monovalent or bivalent) vaccination at the time of chart review.

#### 3.4.1. Chart Review 1 Findings

Charts were randomly selected for CR1 based on adherence to the following criteria: the patient had had an office visit before the intervention period (between 1 October 2022 and 28 February 2023), was 18 years or older, and had a signed partnership form in their patient file. Charts that did not fit those criteria were removed and were not included in the study. After chart identification and data collection, charts were de-identified for the purpose of data analysis and reporting.

The chart reviews performed for CR1 determined that 37.5% of individuals were not “up to date” with the vaccines that they were eligible for and that were recommended by the CDC. Only two individual patients were completely “up to date” with all of the vaccines that they were eligible for and that were recommended by the CDC. Only 43 patients received 50% or more of the vaccines recommended to them.

#### 3.4.2. Chart Review 2 Findings

The same charts from CR1 were reviewed in CR2. Between CR1 and CR2, charts were removed following the protocol described above. As with CR1, after data collection, these charts were de-identified for the purpose of analysis and reporting. At the time of the second chart review (CR2), fewer (25% percent) of the individuals were found to be not “up to date” with their vaccines and only one additional individual (for a total of two) was entirely “up to date” on all vaccines. However, the number of individuals who had received 50% or more of the vaccines recommended to them increased from 43 to 106 at the time CR2 was performed. The chart review also revealed an 88% increase in individuals who had received a bivalent COVID-19 booster and a decrease in COVID-19-unvaccinated patients. Other than for COVID-19 vaccines, there was only a slight increase in specific vaccine uptake between CR1 and CR2 (Figure 1).

While the absolute number of vaccinations increased four-fold during the intervention, the number of individuals who were “up to date” with their recommended vaccinations, based on the chart reviews, only increased by 28% on average during the intervention (Table 1). The definition of “up to date” also changed during the period of the intervention, as a new COVID-19 vaccine booster was introduced. As a result, the number of patients who were “up to date” with their COVID-19 vaccination decreased (Table 2A,B).

### 3.5. Vaccine Administration to CEHC Patients

Separately from the chart review, the clinic was able to track the total number of vaccines ordered and administered during 2022 and 2023, for this study. Individuals who were not included in the chart review (and thus were likely to be receiving vaccines for the first time at CEHC) were asked about vaccination status and vaccinated as per clinic protocol as described above in “Clinic Workflow Changes”. Vaccine administration increased significantly during the intervention period (2023), compared to the previous year (Figure 2). The most significant changes were noted in the following areas: Tdap vaccines (529% increase; 66 doses in 2022 vs. 349 doses in 2023), hepatitis B vaccines (897% increase; 36 doses in 2022 vs. 323 doses in 2023), and the overall vaccines administered across patients visiting the clinic (414% increase; 356 doses in 2022 vs. 1473 doses in 2023) (Figure 3).

During the period from April to August 2023 (when HCP educational sessions were taking place), an average of 133 vaccines were administered each month, compared to an average of 24 during the same period in 2022. CEHC was provided with access to the varicella vaccine by the RI Department of Health in 2023, and over 100 doses were administered during the April to August intervention period. Other routine immunizations that increased during the intervention included hepatitis B, HPV, MMR, Tdap, and pneumococcal pneumonia vaccines.

These improvements were likely attributable to workflow modifications, which allowed for the timely identification and administration of vaccines during patient visits. However, it is also possible that reinforcement of vaccine knowledge through the “Lunch-and-Learn” education sessions gave HCPs more confidence to discuss vaccination schedules during patient visits and to reinforce the importance of being up to date with vaccines.

**Figure 3 vaccines-14-00002-f003:**
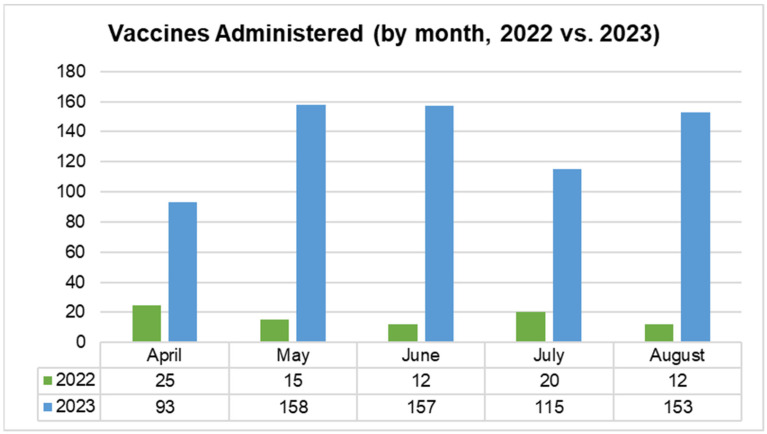
Vaccine administration by month (2022 vs. 2023). This time-based bar chart compares the monthly number of CDC-recommended vaccines administered in two adjacent years, 2022 and 2023. Only the months of April to August were included because that is the period when HCP educational sessions took place, in August 2023. There is a marked upwards trend across all included months.

### 3.6. Survey Results

A total of 36 participants completed pre- and post-educational session surveys, yielding a total of 72 completed surveys. Several measures of vaccination knowledge improved. For example, the proportion of patients correctly identifying the recommended age group for HPV vaccination increased significantly (increases of 76% and 33% in choosing the correct age groups “boys and girls ages 9–26,” and “adults ages 27–45,”, respectively). Additionally, the number of participants who correctly identified a link between cervical cancer prevention and the HPV vaccine increased (17 in the pre-survey, 28 in the post-survey) and more individuals reported that they were willing to vaccinate their children against HPV (18 in the pre-survey, 28 in the post-survey).

The proportion of participants that correctly identified groups at a higher risk of severe COVID-19 infection also increased after the educational intervention. Additionally, survey respondents expressed an increase in willingness to receive the COVID-19 vaccine (an 80% increase between pre- and post-survey values) and to recommend it to others in post-survey responses: a 7% increase for “to the elderly,” a 21% increase in “to my family,” a 41% increase in “to children,” and a 17% increase in “to my friends/colleagues/neighbors”. There was a 100% decrease in respondents saying they do not recommend it between the pre- and post-survey.

Both the pre- and post-surveys showed that patients trust the Navegantes and HCPs at CEHC at a higher rate than other health authorities or trustworthy individuals in patients’ lives such as parents, siblings, close friends, and neighbors (see Figure 4). Patients showed they had less trust in celebrities, athletes, and religious leaders compared to the healthcare team at CEHC. Additionally, survey respondents indicated that discussions with CEHC’s Navegantes or HCPs, or discussions in a healthy lifestyle education class, increased their trust in vaccines. While the sample size of the survey is small, this finding reinforces the hypothesis that CEHC is a highly trusted community organization and source of health information.

### 3.7. Staff Education and Feedback

The “Lunch-and-Learn” sessions were well-received, with an average attendance of seven staff members per session. Staff feedback highlighted an improved confidence in discussing vaccines with patients. Staff survey results demonstrated an increase in knowledge of vaccine guidelines and confidence in administering vaccines. The sample size for staff surveys was small (*n* = 5 for pre-survey, *n* = 4 for post-survey), but findings from these surveys indicated an increase in fact-based knowledge around vaccines, including the correct identification of which vaccines help prevent each infectious disease (Appendix A), the link between HPV infection and cervical cancer, and which higher risk patient populations should receive specific vaccines.

Moreover, the practice and implementation of the AIMS method for discussing vaccines with patients helped to reinforce the workflow modifications and teachings from the “Lunch-and-Learn” sessions. Staff reported higher levels of confidence and comfort when discussing the importance of vaccination with vaccine-hesitant patients in the months following the trainings.

## 4. Discussion

Understanding whether it is patient’s access to vaccines or patient’s vaccine confidence, or both, that are impacting vaccination rates is critical to improving vaccine uptake interventions in the United States, especially among vulnerable populations such as the undocumented, uninsured, and immigrant populations living in urban and rural areas. Vaccine-preventable outbreaks, including measles outbreaks and COVID-19 transmission at the community level, disproportionately affect immigrants who are under-vaccinated [14]. Outbreaks of vaccine-preventable diseases can be severe, affecting thousands of families, as has been observed with measles in New York City [15]. Undocumented, immigrant, and uninsured populations in the United States have been especially hard-hit by outbreaks of infectious diseases such as hepatitis B, measles, mumps, and meningitis. Rates of routinely recommended vaccinations among uninsured adults are significantly lower compared to those who are insured and are as low as 16.6% for hepatitis A, 14.4% for influenza, and 20.9% for HPV, which causes cervical cancer. Vaccination rates have been further suppressed by limited access to care during the COVID-19 pandemic and fears of vaccinations “becoming a public charge” by utilizing government-funded health promotion programs during previous administrations [16,17,18].

CEHC has previously published studies evaluating vaccination and knowledge, attitudes, and practices (KAP), including willingness to participate in vaccination, specifically related to the HPV vaccination, in 2013 [19]. In the previous study, 100 clinic patients were recruited for the study during routine clinic visits. When compared to the non-Hispanic study participants in the intervention, Hispanic study participants also had more limited knowledge of HPV and cervical cancer than non-Hispanics, as has previously been reported. Hispanic survey participants were also less likely to know where to get tested for cervical cancer and STIs, and less knowledgeable on HPV vaccines and the methods for protection against STIs. These findings were consistent with published studies and reflected a need for increased education of the patient population, especially since Latina women are at increased risk for cervical cancer [20,21]. Following this KAP intervention, CEHC conducted a HPV vaccination campaign, “Entre Mujeres” which was extremely successful, causing a 60-fold increase in the number of women receiving HPV vaccinations at CEHC [22].

While the Oportuna Vacuna study did not demonstrate a direct link between the AIMS approach and education sessions and the patient vaccine uptake, the HCP surveys indicated that HCPs had increased confidence in administering, and knowledge about, vaccines. When CEHC combined HCP training on the AIMS method with improved vaccine workflows, more individual patients were vaccinated. Implementation of these methods may improve overall vaccination rates in ambulatory care settings. It is certainly possible that HCPs were more confident when discussing vaccinations with patients at various stages of their visits at the clinic, which improved their ability to complete the vaccination plans that were established by improved workflows for each patient. AIMS training, vaccine education sessions, and workflow modifications are all interventions that can be implemented in other clinics serving vulnerable under-vaccinated populations.

There are limitations to this study. The total number of participants in the survey was small due to concern about inconvenience for patients and because of the decision to prioritize vaccination over surveys. The small sample size may skew the results of the survey. In addition, study subjects were clinic patients, thus their answers may have been influenced by interactions with their healthcare providers at the clinic. Furthermore, study participants may have more positive opinions about vaccination due to their active interest in attending clinic activities to improve their health. They may not be representative of a larger population of uninsured immigrants that do not receive care at free clinics like CEHC. Future studies might consider including a more diverse pool of patients.

## 5. Conclusions

This study demonstrated that providing educational sessions on vaccines to HCPs and patients, while streamlining vaccination procedures in an ambulatory setting, may successfully improve vaccine knowledge and confidence and vaccine uptake. Increasing patient acceptance and HCP confidence concerning vaccines may have similar effects on vaccine uptake in other clinics serving low-income, Spanish-speaking immigrant patients.

However, this project took place in 2023, prior to a significant change of leadership at the United States Health and Human Services (HHS) and the Center for Disease Control (CDC). While vaccine recommendations have changed, it is important to recall that the value of education and training for healthcare providers to support vaccine acceptance by patients has not changed. Vaccination protects families and communities; therefore, work to improve vaccine uptake in all communities, including low income, uninsured, and immigrant communities, should continue. Such efforts, organized by trusted community-based organizations like CEHC are vital to maintaining continuity of care and continued access to health resources.

## Figures and Tables

**Figure 1 vaccines-14-00002-f001:**
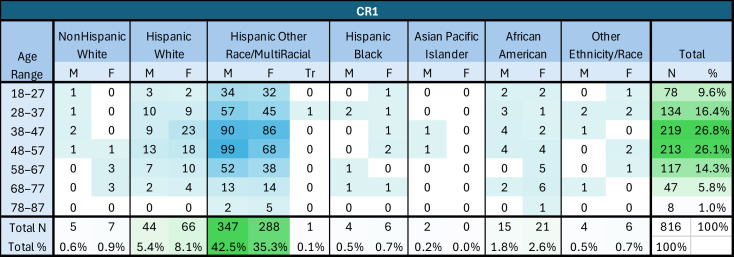
A heat map view of CEHC patient demographics in the initial chart review cohort. This heat map overlaid on a frequency table shows the cohort participants’ age range and ethnicity/race. The increased intensity of the blue color indicates increased frequency among demographic categories. The increased intensity of the green color indicates increased frequency and relative frequency in the totals. M indicates male gender. F indicates female gender. Tr indicates that a survey participant marked a gender other than male or female.

**Figure 2 vaccines-14-00002-f002:**
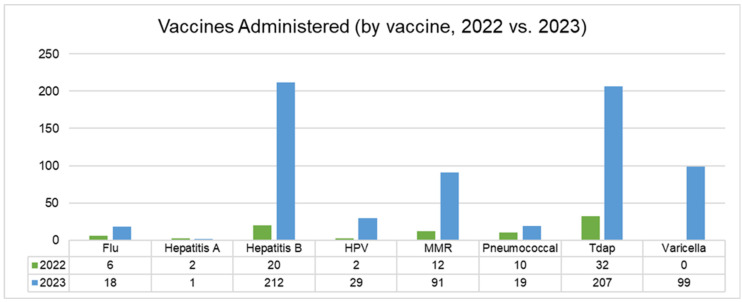
Vaccine administration by vaccine (2022 vs. 2023). This is a time-based bar chart that compares the number of CDC-recommended vaccines administered between 2022 and 2023. There is a marked upwards trend across all vaccines listed, except for hepatitis A.

**Figure 4 vaccines-14-00002-f004:**
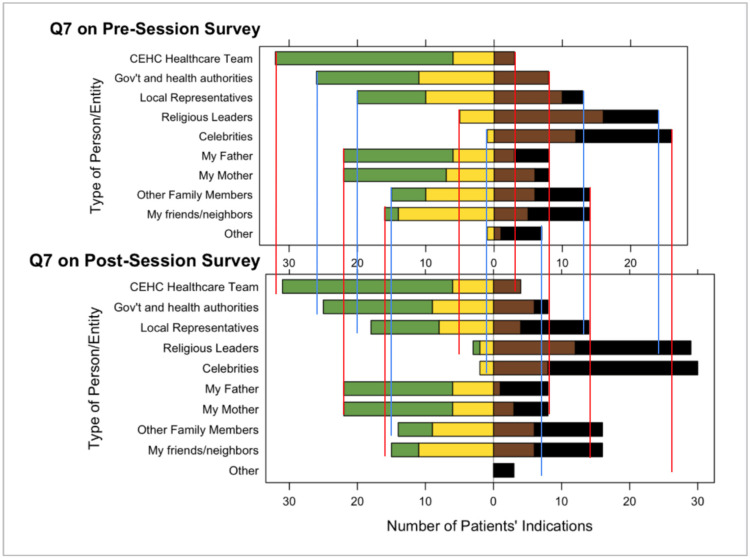
Survey question 7 in pre- vs. post-educational session surveys. This figure provides a visualization of the Likert-scaled survey question 7, using a diverging stacked bar chart centered on 0. It compares responses before and after the educational session for 36 participants when asked the following question: indicate how often you follow the recommendations of the people/entities below regarding vaccination. Green represents the number of participants who responded Always. Yellow represents the number who responded Often. Brown represents the number who responded Sometimes and black represents the number who responded Never. The red and blue lines are included for ease of comparison between the pre-session survey and the post-session survey.

**Table 1 vaccines-14-00002-t001:** Chart review: number of patients that were “up to date” by vaccine.

Vaccine	CR1	CR2
Flu (2022–2023 batch)	309	315
Tdap	243	308
Pneumococcal	132	158
Hep B	56	119
MMR	46	71
Varicella	30	56
HPV	30	38

Comparison of the number of patients that were “up to date,” for each vaccine. “Up to date” at the time of the chart review (1 or 2) was defined as having received all of the vaccines that the CDC recommended in 2022–2023.

**Table 2 vaccines-14-00002-t002:** (**A**)—Chart review: COVID-19 vaccine uptake. (**B**)—Chart review: COVID-19 vaccine uptake.

(A)					
	CR1 (*n* = 816)		CR2 (*n* = 709)		
COVID-19 Status	# of Pts	% of Total Pts	# of Pts	% of Total Pts	Difference in Percentage Points
Primary series still in progress *	18	2.21%	9	1.27%	−0.94%
Completed primary series	664	81.37%	607	85.61%	4.24%
Received monovalent booster	311	38.11%	297	41.89%	3.78%
Received bivalent booster (s)	100	12.25%	188	26.52%	14.26%
“Up to date” on series/booster as defined by CDC *	105	12.87%	181	25.53%	12.66%
Unvaccinated (0 doses)	33	4.04%	12	1.69%	−2.35%
Missing documentation/unconfirmed vaccination status	96	11.76%	67	9.45%	−2.31%
(**B**)					
**COVID-19 Status**			**CR1**		**CR2**
Completed primary series			664		607
Received monovalent booster (s)			311		297
Received bivalent booster (s)			100		188

(**A**). A comparison of the first and second chart reviews by the COVID-19 vaccination status of the patients, including primary series vaccination and monovalent booster vaccination, after the introduction of the new bivalent booster. Here the symbol * indicates that monovalent COVID-19 vaccines lost FDA approval for use in April 2023, meaning that the vaccine cadence was updated by health authorities. Some categories in the table overlap; for example, those who received a monovalent booster could have also completed a primary series vaccination. (**B**). Comparison of the first and second chart reviews by COVID-19 vaccination status following the introduction of the new bivalent booster.

## Data Availability

The data presented in this study are available upon request from the corresponding authors in compliance with HIPAA and all applicable privacy laws.

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
