# Peer review of "Oportuna Vacuna: A Prospective Study of Vaccine Confidence and Vaccine Uptake in a Low-Income, Spanish-Speaking Rhode Island Population in the Post-Pandemic Era"

_vaccines, 2025, doi:10.3390/vaccines14010002_

Round 1

Reviewer 1 Report

Comments and Suggestions for Authors

This is an interesting study presented on a vulnerable social group. I appreciated the way the study was structured, how the samples were selected, and the attention paid to capturing all the necessary nuances. It's sad, of course, that in the 21st century so many people know nothing about vaccinations or have negative attitudes toward them. A lack of knowledge is likely to be the cause. I was very pleased with the final result of the extensive work done and the resulting increased vaccination rates among the population. Despite this, there are several issues that do not detract from the overall positive impression of this manuscript.

1.Line 253. The authors indicate that there were originally 816 maps, of which 709 remained participants in this study. Am I correct in understanding this? So, there was virtually no migration? Please clarify.

2. Lines 347 – 349. “However, it is also possible that reinforcement of vaccine knowledge through the “Lunch 347 and Learn” education sessions gave HCPs more confidence to discuss vaccination schedules during patient visits and to reinforce the importance of being up to date with vaccine”. It's not entirely clear what we're talking about. If medical staff have a strict vaccination schedule, how can conversations outside of working hours affect that? Or if there's no vaccination schedule and vaccines are administered sporadically after patients seek medical care, then what kind of system is this? Please clarify. Because vaccinations are based on the patient's condition, while vaccine administration requires an initial patient examination for contraindications. It's not entirely clear.

3.Lines 447-451. So, it turns out that not only patients but also healthcare workers are improving their vaccine knowledge? Please briefly describe (in any convenient section of the article) whether healthcare workers receive training in immunology/virology/microbiology as part of their primary medical education. Perhaps this is done differently in different countries, which is why some misunderstandings arise when reading the article.

Author Response

This is an interesting study presented on a vulnerable social group. I appreciated the way the study was structured, how the samples were selected, and the attention paid to capturing all the necessary nuances. It's sad, of course, that in the 21st century so many people know nothing about vaccinations or have negative attitudes toward them. A lack of knowledge is likely to be the cause. I was very pleased with the final result of the extensive work done and the resulting increased vaccination rates among the population. Despite this, there are several issues that do not detract from the overall positive impression of this manuscript.

1.Line 253. The authors indicate that there were originally 816 maps, of which 709 remained participants in this study. Am I correct in understanding this? So, there was virtually no migration? Please clarify.

Thank you for your request for clarification. The Heat Map in figure 1 only shows the demographics at the time of the initial chart review – at which time 816 charts were evaluated. Line 236 to 237 provides some additional clarification – that the number of charts reviewed at the 2nd chart review (CR2) was 709. We did not provide a demographics figure for the 2nd chart review.

“816 charts were randomly selected for the initial chart review (CR1); 709 of these individuals were still active patients at CEHC at the time of the post-intervention chart review (CR2).”

Patients were lost to follow up during the period of the study and removed from the chart review. Exclusion criteria are in section 2.2 Chart Review, lines 128-139.

We would like to clarify that the only charts that were removed were of patients who fit certain criteria that would make them ineligible to receive care at CEHC including, but not limited to: becoming insured, moving states, or passing away. Patients were also removed if staff were unable to reach them following the clinic’s outreach protocol which includes phone calls and sending physical mail. If a patient did not fit the removal criteria, they were kept in the chart review population, which is why there was little movement. While the patient population is transient, they are largely ineligible for insurance, move seasonally or for a few weeks to a few months rather than permanently.

2. Lines 347 – 349. “However, it is also possible that reinforcement of vaccine knowledge through the “Lunch 347 and Learn” education sessions gave HCPs more confidence to discuss vaccination schedules during patient visits and to reinforce the importance of being up to date with vaccine”. It's not entirely clear what we're talking about. If medical staff have a strict vaccination schedule, how can conversations outside of working hours affect that? Or if there's no vaccination schedule and vaccines are administered sporadically after patients seek medical care, then what kind of system is this? Please clarify. Because vaccinations are based on the patient's condition, while vaccine administration requires an initial patient examination for contraindications. It's not entirely clear.

Thank you again for this request for clarification. We based our vaccine protocols (who to vaccinate when) on the guidelines that were active at the time of the study. Since the guidelines have changed dramatically, we have now included the full Adult Schedule as it was established by CDC in 2023, in the supplemental data as a new appendix and updated the references to include a source for this information(Appendix C).

The reference to the source of the 2023 guidelines follows:

Neil Murthy, A. Patricia Wodi, Sybil Cineas, et al. Recommended Adult Immunization Schedule, United States, 2023. Ann Intern Med.2023;176:367-380. doi:10.7326/M23-0041

Patients were vaccinated at their appointments and if subsequent doses are required, the follow-up appointments were scheduled during the checkout process during their first vaccination visit.

 The protocols are explained under section 2.5 Clinic Workflow Changes (lines 208 – 224). The following information was added to line 210 for clarification:

Vaccines were administered by Navegantes and nursing staff using a pre-approved clinic protocol (based on the 2023 CDC recommendations, Appendix C(ref)), reducing the need for individual review and approval by supervisory HCP.

3.Lines 447-451. So, it turns out that not only patients but also healthcare workers are improving their vaccine knowledge? Please briefly describe (in any convenient section of the article) whether healthcare workers receive training in immunology/virology/microbiology as part of their primary medical education. Perhaps this is done differently in different countries, which is why some misunderstandings arise when reading the article.

 Yes! HCW, and the Navegantes in particular, improved their knowledge about vaccines during the project, based on the survey reviews. We believe that this was key to the success of the project. While licensed medical providers (MD, PA, RN) learn about vaccines during their formal clinical education, Navegantes and other non-medical provider HCWs, only really learn the basics about vaccines and vaccine-preventable diseases during formal trainings and education sessions organized ‘on the job’ and provided by the. The Lunch & Learn materials covered the most basic, high-level concepts of immunology/virology, as their position as a Navegante does not require relaying in-depth information and details about those topics to patients.

See the text explaining the above in lines following line 185 after “… and current CDC guidelines,” (section 2.4, paragraph 2).

Reviewer 2 Report

Comments and Suggestions for Authors

The authors conducted a survey on vaccination attitudes in low-income, predominantly Spanish-speaking populations living in the USA. The study adopted a prospective pre-post design. Health data was treated with the utmost confidentiality, scrupulously adhering to the limits of informed consent. The intervention included training for operators and users.

  1. The training methodology adopted criteria of considerable interest. The authors may consider adding more detailed information on this issue as a supplement.
  2. The experiment was conducted in 2022-2023. We know that the U.S. government's attitude toward recent immigrants has changed recently. Could the authors advise readers if they believe their experiment should and could be continued?
  3. Although the authors contacted many people, only a few participants completed pre- and post-educational session surveys. Beyond this limitation of the study, could the authors perhaps tell us what might be causing such low participation?
  4. The Discussion does not include a section on limitations. It should include, among other things, the fact that the research was conducted at a single center and the authors should tell us how representative it is of the entire population; the fact that few people completed the pre-post observation; and the fact that vaccination schedules and observation conditions changed over the course of the observation.

Author Response

The authors conducted a survey on vaccination attitudes in low-income, predominantly Spanish-speaking populations living in the USA. The study adopted a prospective pre-post design. Health data was treated with the utmost confidentiality, scrupulously adhering to the limits of informed consent. The intervention included training for operators and users.

  1. The training methodology adopted criteria of considerable interest. The authors may consider adding more detailed information on this issue as a supplement.

We appreciate the valuable feedback about this matter although we are not exactly sure what methodology is referenced. If the reviewer means methods for improving vaccine acceptance, the “AIMS” vaccine acceptance training methodology was described in referenced material. Please see line 190 (where reference 11 is mentioned) to line 204. This approach was provided to clinicians and the Navegantes were taught the method during the lunch and learn sessions.

  1. The experiment was conducted in 2022-2023. We know that the U.S. government's attitude toward recent immigrants has changed recently. Could the authors advise readers if they believe their experiment should and could be continued?

We agree with the reviewer and do believe that projects like this should continue even in the wake of the federal government’s changed position on immigrants and immigration more widely. We added the following text. 

However, this project took place in 2023, prior to a significant change of leadership at the United States Health and Human Services (HHS) and the Center for Disease control (CDC). While vaccine recommendations have changed, it is important to recall that the value of education and training for healthcare providers to support vaccine acceptance by patients has not changed. Vaccination protects families and communities, and therefore, work to improve vaccine uptake in all communities including low income, uninsured, and immigrant communities, should continue. Such efforts, organized by trusted community-based organizations like CEHC are vital to maintaining continuity of care and continued access to health resources.

We added this text to the next to final paragraph at line 477.

3. Although the authors contacted many people, only a few participants completed pre- and post-educational session surveys. Beyond this limitation of the study, could the authors perhaps tell us what might be causing such low participation?

We agree and would like to provide clarification about participant survey ration. Surveys were only given to individual patients who attended healthy lifestyle education classes in either group or one-on-one settings, and who were given a vaccine education session. This decision was made in order to not inconvenience patients during their routine visits to the clinic and to ensure the information was delivered completely and that the participants had an appropriate amount of time to ask questions about the survey.

  1. The Discussion does not include a section on limitations. It should include, among other things, the fact that the research was conducted at a single center and the authors should tell us how representative it is of the entire population; the fact that few people completed the pre-post observation; and the fact that vaccination schedules and observation conditions changed over the course of the observation.

We agree and we have added information about the limits of the study and changes to be made if it is replicated at line 462.

There are limitations to this study. The total number of participants in the survey was small due to concern about inconvenience for patients (prioritizing vaccination over surveys) The small sample size may skew the results of the survey. In addition, study subjects were clinic patients, thus their answers may have been influenced by interactions with their care providers. Furthermore, study participants may have more positive opinions about vaccination due to their active interest in attending clinic activities to improve their health. They may not be representative of a larger population of uninsured immigrants that do not attend free healthcare clinics. Future studies might consider including a more diverse pool of patients.

Reviewer 3 Report

Comments and Suggestions for Authors

The 4th citation DOI does not lead to the paper. It seems there is an ordinary web address but not a DOI for the paper. Please check that. I think you could discuss the issue further by reading more about this topic and citing more studies. 

Also, I think the Discussion section should include Limitations on your study conclusions. You could mention other possible reasons for increasing uptake of vaccines, and also include any Cost-related discussions that can be relevant to the issue of access to vaccines. There is a lot of literature that covers this aspect of accessibility including direct and indirect costs related to visits of a health center for vaccination. 

Author Response

The 4th citation DOI does not lead to the paper. It seems there is an ordinary web address but not a DOI for the paper. Please check that. I think you could discuss the issue further by reading more about this topic and citing more studies. 

Thank you for identifying the problem with the citation. This was a sloppy copy paste! that was our fault - There was an ‘external link icon’ in the citation at the end of that DOI. Please see the corrected reference below, which is replaced, under the reference section, reference #4.

Santoli JM, Lindley MC, DeSilva MB, et al. Effects of the COVID-19 Pandemic on Routine Pediatric Vaccine Ordering and Administration — United States, 2020. MMWR Morb Mortal Wkly Rep 2020, 69, 591–593. DOI: http://dx.doi.org/10.15585/mmwr.mm6919e2

Also, I think the Discussion section should include Limitations on your study conclusions. You could mention other possible reasons for increasing uptake of vaccines and also include any Cost-related discussions that can be relevant to the issue of access to vaccines. There is a lot of literature that covers this aspect of accessibility including direct and indirect costs related to visits of a health center for vaccination. 

We agree that a discussion of cost-effectiveness would be an important addition however, it would require additional information about the activities at the clinic (such as the cost of providing vaccines) that we did not collect in 2023. This is an excellent topic for future research at the clinic and we do have experience calculating cost-savings for the free care that is provided at CEHC (we published this previously in an article on the “CHEER” walk in clinic and “Bridging the Gap” (Hindocha, et al. (2018). Bridging the [Health Equity] Gap at a Free Clinic for Uninsured Residents of Rhode Island. Rhode Island medical journal (2013)101(9), 27–31.) and Barry, K., et al. (2019). Four Years of CHEER: Cost and QALY Savings of a Free Nurse-run Walk-in Clinic Serving an Uninsured, Predominantly Spanish-speaking Immigrant Population in Providence. Journal of Health Care for the Poor and Underserved 30(2), 806-819. https://dx.doi.org/10.1353/hpu.2019.0057.

For this reason, we have elected not to introduce cost-effectiveness and will keep this excellent suggestion in mind for future studies.

We also agree that we need to add information about the limitations of the study, and we have added information at line 462:

There are limitations to this study. The total number of participants in the survey was small due to concern about inconvenience for patients (prioritizing vaccination over surveys) The small sample size may skew the results of the survey. In addition, study subjects were clinic patients, thus their answers may have been influenced by interactions with their care providers. Furthermore, study participants may have more positive opinions about vaccination due to their active interest in attending clinic activities to improve their health. They may not be representative of a larger population of uninsured immigrants that do not attend free healthcare clinics. Future studies might consider including a more diverse pool of patients.

Round 2

Reviewer 3 Report

Comments and Suggestions for Authors

Thank you. The corrections done by the team are enough. Best of luck to your team. You need a "." at the end of the sentence here: There are limitations to this study. The total number of participants in the survey was small, due to concern about inconvenience for patients (prioritizing vaccination over surveys) The small sample size may skew the results of the survey.

I also suggest you change the writing and add the information within the parentheses ( ) without the parentheses themselves. 

Author Response

Thank you very much for finding this error and requesting a re-write of the sentence. 
The sentence now reads: 
The total number of participants in the survey was small, due to concern about inconvenience for patients, and because of the decision to prioritize vaccination over surveys.